# Estimating the Burden of Stroke: Two-Year Societal Costs and Generic Health-Related Quality of Life of the Restore4Stroke Cohort

**DOI:** 10.3390/ijerph191711110

**Published:** 2022-09-05

**Authors:** Ghislaine van Mastrigt, Caroline van Heugten, Anne Visser-Meily, Leonarda Bremmers, Silvia Evers

**Affiliations:** 1Department of Health Services Research, CAPHRI School for Public Health and Primary Care, Faculty of Health, Medicine and Life Sciences, Maastricht University, 6200 MD Maastricht, The Netherlands; 2MHeNS, School for Mental Health & Neuroscience, Department of Psychiatry & Psychology, Faculty of Health Medicine and Life Sciences, Maastricht University, 6229 ER Maastricht, The Netherlands; 3Department of Neuropsychology & Psychopharmacology, Faculty of Psychology & Neuroscience, Maastricht University, 6229 ER Maastricht, The Netherlands; 4Department of Rehabilitation, Physical Therapy Science and Sports, Brain Center, University Medical Center Utrecht, 3584 CX Utrecht, The Netherlands; 5Erasmus Centre for Health Economics Rotterdam (EsCHER), Erasmus University, 3062 PA Rotterdam, The Netherlands; 6Trimbos Institute, Netherlands Institute of Mental Health and Addiction Utrecht, 3521 VS Utrechtcity, The Netherlands

**Keywords:** longitudinal cohort, societal costs, quality of life, stroke, burden of disease, costs-of-illness

## Abstract

(1) Background: This study aimed to investigate two-year societal costs and generic health-related quality of life (QoL) using a bottom-up approach for the Restore4Stroke Cohort. (2) Methods: Adult post-stroke patients were recruited from stroke units throughout the Netherlands. The societal costs were calculated for healthcare and non-healthcare costs in the first two years after stroke. The QoL was measured using EQ-5D-3L. The differences between (sub)groups over time were investigated using a non-parametric bootstrapping method. (3) Results: A total of 344 post-stroke patients were included. The total two-year societal costs of a post-stroke were EUR 47,502 (standard deviation (SD = EUR 2628)). The healthcare costs decreased by two thirds in the second year −EUR 14,277 (95% confidence interval −EUR 17,319, −EUR 11,236). In the second year, over 50% of the total societal costs were connected to non-healthcare costs (such as informal care, paid help, and the inability to perform unpaid labor). Sensitivity analyses confirmed the importance of including non-healthcare costs for long-term follow-up. The subgroup analyses showed that patients who did not return home after discharge, and those with moderate to severe stroke symptoms, incurred significantly more costs compared to patients who went directly home and those who reported fewer symptoms. QoL was stable over time except for the stroke patients over 75 years of age, where a significant and clinically meaningful decrease in QoL over time was observed. (4) Conclusions: The non-healthcare costs have a substantial impact on the first- and second-year total societal costs post-stroke. Therefore, to obtain a complete picture of all the relevant costs related to a stroke, a societal perspective with a follow-up of at least two years is highly recommended. Additionally, more research is needed to investigate the decline in QoL found in stroke patients above the age of 75 years.

## 1. Introduction

Based on the Global Burden of Diseases Study data, the lifetime risk of stroke from the age of 25 years onward is approximately 25% among both men and women [1], while strokes remain the second leading cause of death worldwide [2]. These global figures are expected to change over the coming decades due to population growth and aging [3]. Moreover, the care for patients with stroke has drastically transformed over the past seven years due to the introduction of reperfusion therapies for ischemic stroke and improved secondary prevention [3]. This has resulted in improved stroke survival rates, and a higher prevalence of chronic stroke [2]. However, The American Heart Association forecasted that, by 2030, almost 4% of US adults will have had a stroke [4].

They also estimated an annual total healthcare cost of USD 30.8 billion in 2016 and 2017 as a result of strokes [5]. Currently, in Western countries, approximately 1.7% to 4% of total healthcare expenditures are stroke-related [6]. In addition to high healthcare costs, the disease also generates non-healthcare costs, such as costs related to (in)formal care provision or productivity losses [5,7,8]. In the first year post stroke, these costs could account for anything from 34% to over 75% of the total societal costs [7,9]. Hence, strokes have a considerable effect on the psychological and emotional well-being of patients, which is also strongly reflected in a reduced health-related quality of life (QoL) [10,11]. The effects of a stroke can interfere with many aspects of daily life [11] and in some cases may render the patient fully dependent on others [12]. Considering both the high (non-) healthcare costs of strokes and their adverse impact on QoL, there is a need to investigate the burden of illness of a stroke in the long term to obtain insights into when costs occur and how a stroke affects QoL.

Building on previous work from the Restore4Stroke study [13], the one-year societal costs and QoL of stroke patients was investigated [7]. However, little is known about the long-term (more than one year) costs and QoL post stroke using a societal perspective and using a bottom-up costing approach [5,8,14,15]. It is hypothesized that the costs of a stroke will peak within the first year and decline over time, while the QoL will increase. However, there is a need for further exploration of long-term resource use [16].

The current study involves a bottom-up Burden of Disease (BoD) study from a societal perspective with a two-year follow-up. More specifically, the aims of the current study were: (1) to estimate the total societal costs two years post stroke; (2) to compare both the healthcare and the non-healthcare costs measured in the first and second-years post-stroke; (3) to measure the impact of strokes on generic health-related QoL at one and two years post-stroke; and (4) to identify relevant stroke subgroups. 

## 2. Materials and Methods

### 2.1. Study Design

The current study was embedded in a prospective, multi-center, observational cohort study entitled ”Restore4Stroke”, which aims to gain insight into care received post-stroke, and the economic consequences of psychosocial care in the first two years post-stroke. The design of this BoD study [13], the short-term follow-up of BoD data [7], and the short- to long-term follow-up of disease-specific quality of life has been published elsewhere [17,18]. The current BoD study reports on two-year follow-up results of societal costs and health-related related QoL. This study was performed according to the Dutch guidelines for economic evaluations in healthcare [19], reported according to the Consolidated Health Economic Evaluation Reporting Standards (CHEERS 2022) guidelines [20] and Strengthening the Reporting of OBservational Studies in Epidemiology (STROBE) [21]. The datasets used and/or analyzed during the current study are available from the corresponding author upon reasonable request.

### 2.2. Setting and Participants

Stroke survivors were recruited from stroke units in six general hospitals throughout the Netherlands: St. Antonius Hospital (Nieuwegein); Diakonessenhuis (Utrecht); Canisius Wilhelmina Hospital (Nijmegen); TweeSteden Hospital (Tilburg); St. Elisabeth Hospital (Tilburg); Catharina Hospital (Eindhoven). Patients were eligible for inclusion if they had a clinically confirmed stroke (either first or recurrent stroke) within the last seven days and were at least 18 years of age. The exclusion criteria were if the patient (1) had any other condition that may be expected to interfere with the study outcomes (e.g., neuromuscular disease), (2) were clinically judged unable to sufficiently understand and complete Dutch questionnaires, (3) were rated physically dependent as defined by a Barthel Index (BI) [22] score of 17 or below, or (4) had existing cognitive decline before the stroke as defined by a score of 1 or higher on the Heteroanamnesis List Cognition [23]. The medical ethics committees of all participating hospitals approved the Restore4Stroke Cohort study and informed consent was obtained from all included patients. Patients were treated according to Dutch guidelines for stroke treatment and rehabilitation [24], including general aftercare at the outpatient clinic of neurology at six to eight weeks post-discharge in addition to regular follow-up for secondary prevention purposes.

### 2.3. Procedures

Patients received information on the study and were asked to give informed consent by the nurse practitioner or the trial nurse during the hospital stay in the first-week post-stroke. After informed consent was obtained, the patients’ characteristics were collected. The generic health-related QoL questionnaires (EuroQoL-5D-3L) [25] and the 11-item cost questionnaires were filled in at two months, six months, one year, and two years post-stroke. At two months and six months, a research assistant visited the stroke patient at home or at the institution where the patient resided at that moment. Questionnaires were sent to the patients in advance and trial nurses assisted with completion. At six months, one year, and two years post-stroke, patients could choose to fill in an online or paper version of cost and QoL questionnaires. All administered questions were checked by trial nurses to avoid missing data. All patients were contacted several times by phone in case of delay or non-response.

### 2.4. Costing

A bottom-up costing approach from a societal perspective using a specifically designed 11-item cost questionnaire was performed (Appendix A). The questionnaire consisted of open questions measuring both healthcare and non-healthcare resource use. Healthcare items were related to the number of general practitioner, specialist, allied health professional, mental healthcare professional and rehabilitation treatment visits, as well as overnight stays in hospitals, rehabilitation clinics, nursing homes, and psychiatric clinics. In addition, information on medication use (duration, amount, and dosages) was collected. Non-healthcare resource use questions were related to expenses incurred via paid home care, informal unpaid care, and the patient’s inability to perform unpaid and paid labor (productivity losses). 

The valuation of the costs was conducted using the updated Dutch Manual for Cost Analysis in Health Care Research [26]. An overview of the healthcare and non-healthcare unit costs was reported and is included in the Appendix A. Costs of medication (prescribed and over-the-counter drugs) were valued according to the market prices in the summer of 2019 (including 6% tax) [27]. To account for uncertainty, the lowest cost price was used to estimate drug costs. The friction cost method was used to calculate productivity costs by calculating production losses confined to the period required to replace a sick employee (85 working days or 12 weeks) [28]. Productivity costs were not calculated for patients over 66 years of age. All costs were valued in 2018 Euros. Discounting was applied for costs after 12 months using a rate of 4% [19].

### 2.5. Quality of Life (QoL) Measurement

The descriptive system of the three-level EuroQoL (EQ-5D-3L) is a widely used and recommended preference-based measure of health [25]. The five items refer to the following dimensions: mobility; self-care; usual activities; pain/discomfort; anxiety/depression. The three-level scale utilized for scoring included: no problems; some problems; extreme problems. The EQ-5D-3L for use in post-stroke patients has shown reasonable validity and reliability. However, in this same study, limitations in responsiveness were observed [29]. The EQ-5D-index or EuroQoL utility score was derived from the EQ-5D-3L using the Dutch value set [30]. The lower the scores on the EuroQoL dimensions or Euroqol utility score, the worse the quality of life. A discount rate of 1.5% was applied in the second year post-stroke, as recommended by Dutch PharmacoEconomics guidelines [19]. Clinically meaningful differences in utility measurements were estimated using 0.5 times the standard deviation on a baseline measurement [31]. 

### 2.6. Handling of Missing Data

When two or more complete assessments were missing from the cost and/or quality of life questionnaires, patients were excluded from the analyses. Diseased patients were always included in the analysis. The EQ-5D-3L scores for diseased patients were set to the maximum ranking for all dimensions (i.e., a score of 3/3/3/3/3) and the costs were defined as zero for all measurements after their reported date of death.

Cost questionnaires were filled in at two months (retrospective for two months), six months (retrospective for four months), and at twelve and twenty-four months (retrospective for six months). In the base case analysis, to calculate the total resource use of the second year (12 to 24 months period of 12 months in total), the 24-month measurement was doubled (twice the reported 6 months of resource use).

### 2.7. Statistics

Multiple imputations were applied to replace missing cost and QoL data using the following predictors: gender; marital status; age; treatment location; severity of stroke (National Institutes of Health Stroke Scale (NIHSS) rating) [32,33]. The resource use and costs were reported for the total follow-up period (24 months) and two study periods (0–12 and 12–24 months post-stroke). These data were reported in mean, medians, and standard deviations as well as in mean differences and 95% confidence intervals (95%CIs). Since cost data are generally known to be skewed, we used non-parametric bootstrapping to estimate the uncertainty. Different sets of replication runs were tested, and 1000 replications will result in stable outcomes [19]. Statistical differences between the groups were analyzed using the same bootstrapping technique (Excel 2016) as in the cost analyses. The scores on the five dimensions of the EuroQoL and Dutch utilities were reported as mean, standard deviation for the total group, by gender, age groups (<65, 65–74 and >74 years of age), and stroke severity (NIHSS) at every measurement point. All other analyses were performed using IBM SPSS Statistics version 25.

### 2.8. Sensitivity and Subgroup Analyses

Four sensitivity analyses were performed to test for differences between follow-up measurements or follow-up periods. In the first sensitivity analysis, the subject of investigation was the extrapolation of the costs from a 12 to 18-month period. In the base case analyses, to estimate the total costs for the second year after stroke, the total costs at 24 months (retrospective measurement for 6 months) were multiplied by two. In the sensitivity analysis, the costs at 12 months (retrospective measurement for 6 months) and the costs at 24 months were added to estimate the costs of the same period. In the second sensitivity analysis, the societal perspective (taking into account both health and non-healthcare costs) was compared to the healthcare perspective (only using healthcare costs). In the two other sensitivity analyses, the base case Dutch tariffs [30] for the valuation of utilities were replaced by the UK tariffs [34] at 12 and 24 months, respectively. 

Both generic health-related quality of life and the total societal costs were investigated in seven subgroups. These groups were gender (male/female), age (65−/65+), education level (low/high), stroke type (infarction/hemorrhage), recurrent stroke (yes/no), home discharge (yes/no), and severity of stroke (NHISS 0–4/>5). 

## 3. Results

### 3.1. Patient Characteristics

A total of 78% (*n* = 344) of the initial Restore4Stroke Cohort (*n* = 395) was included in the current study. A total of 35 patients (8.9%) dropped out and another 16 patients (4.4%) were excluded as they had two or more assessments of costs and/or QoL measurements missing. More details on the missing data are reported in Figure 1.

Table 1 shows the background characteristics of the post-stroke patients. The mean age at stroke onset was 66.7 years, and 35.5% of the patients were female. One-third of the patients were not in a relationship and had completed higher education at the start of the study. Over 75% of the patients reported no or minor stroke symptoms. A total of 97 patients (28.2%) did not go home after their discharge from the stroke unit. These patients were either admitted to a rehabilitation center (*n* = 48, 14.0%) or to a nursing home (*n* = 49, 14.2%).

### 3.2. The Total Societal Costs over Two Years Post-Stroke

The average total societal cost for patients two years post-stroke was EUR 47,502 (SD = EUR 2628). The mean healthcare cost was estimated to be EUR 27,159 (SD = EUR 1611) and the mean non-healthcare costs were on average EUR 20,330 (SD = EUR 1603) per patient. Over 80% (EUR 21,829) of the healthcare costs were related to outpatient rehabilitation and inpatient hospital and rehabilitation stay. The other 20% of the healthcare costs were related to the remaining healthcare cost categories, such as costs of medication, general practitioner, and specialist visits. 60% of the non-healthcare costs in the two years after a stroke are related to paid home care, informal care, and the cost of productivity losses. The other 40% of costs (EUR 8284) are due to the patients’ inability to perform unpaid labor. For further details on resource use and costs in the 24 months post-stroke, see Table 2.

### 3.3. Healthcare and Non-Healthcare Costs in the First and Second-Year Costs after Stroke 

A significant difference was reported in societal costs between the first year and the second year after a stroke (mean = −EUR 16,703, 95%CI = −EUR 21,243, −EUR 12,039) (see Table 3). The total costs in the second year post-stroke (mean = EUR 15,383, SD = EUR 1526) were on average one third lower compared to the total costs reported in the first year (mean = EUR 32,085, SD = EUR 1743). The total healthcare costs were also significantly lower in the second compared to the first year after a stroke (mean = −EUR 14,277, 95%CI = −EUR 17,319, −EUR 11,236). This was mainly caused by significantly lower costs in three major cost categories: outpatient rehabilitation, inpatient hospital stay, and inpatient rehabilitation stay. Additionally, stroke patients reported fewer visits to general practitioners and allied health and mental healthcare professionals in the second year after stroke compared to the first year. The non-healthcare costs were lower in the second year, but this difference was not statistically significant (mean = −EUR 2426, 95%CI = −EUR 5079, EUR 246). In the second-year post-stroke, significantly higher costs were reported for the inability to perform unpaid labor. In the first year, the estimated costs due to productivity losses were higher compared to the second year. This was caused by the method used for the calculation of the productivity losses (friction cost method).

### 3.4. Generic Health-Related Quality of Life 2 Years Post-Stroke

The scores with respect to the five dimensions of the EQ-5D-3L were comparable at 12 months post-stroke compared to 24 months post-stroke. This also accounts for the utilities estimated using the Dutch tariffs for either group. When the patients were divided into subgroups for “gender” (male/female) and “severity of stroke” (NIHSS categories 0, 1–4, 5–12 and ≥13), no significant differences in utilities were reported in time. Only patients over 75 years of age had a significant decrease and clinically meaningful lower utility score at 24 months compared to 12 months post-stroke (mean = −0.14, 95%CI = −0.23, −0.05). For details, see Table 4.

### 3.5. Sensitivity Analyses & Subgroup Analyses

The sensitivity analysis shows that the cost extrapolation method in the second year after stroke did not change the base case results. In another sensitivity analysis, it was found that societal costs were significantly higher compared to healthcare costs (mean = −EUR 15,382, 95%CI = −EUR 21,244, −EUR 9785).

When the Dutch utilities were compared with UK tariffs at one year, no significant difference was reported (mean = −0.0506, 95%CI = −0.1104, 0.0081). At 24 months, however, significantly different utilities were observed (mean = −0.0644, 95%CI = −0.1122, −0.0167). However, this difference could not be defined as clinically meaningful. For details, see Table 5.

At 12 months utility, significant differences were observed in the subgroups “age” (65+/65−) and “discharge at home” (Yes/No), the 24 months utility measurements, and for the subgroup “stroke type” (Infarction/Hemorrhage). However, only the subgroup “stroke type” could also be defined as clinically meaningful. In other words, in the hemorrhage subgroup, the utilities were lower compared to the infarction group. After two years, the societal costs were significantly higher when the patients were transferred to rehabilitation clinics or geriatric rehabilitation centers (*n* = 97) compared to home discharge (Table 5). Patients discharged to the geriatric rehabilitation centers (*n* = 49) incurred twice as many costs as patients who went home (*n* = 247) or were transferred to rehabilitation clinics (*n* = 48) in the second year (Figure 2A). However, this difference was not observed in the first year post-stroke. The costs of patients with moderate to severe stroke symptoms were on average twice as high as those with fewer symptoms during a follow-up (Table 5, Figure 2B). 

## 4. Discussion

Examining the recent period, this is the first prospective burden of disease study performed using a societal perspective and a bottom-up costing approach with a two-year follow-up. The average societal cost for a stroke patient after 24 months was EUR 47,502 (SD = EUR 2628). A decline in societal costs in the second year was observed (mean = −EUR 16,703, 95% CI = −EUR 21,243, −EUR 12,039). However, the non-healthcare costs, such as (in)formal care costs and the cost of not being able to perform unpaid labor, have a substantial impact on the total costs in the second year after a stroke. The sensitivity analyses confirmed the importance of including non-healthcare costs for the long-term follow-up. The subgroup analyses showed that the patients who were transferred to rehabilitation clinics or geriatric centers, and those with moderate to severe stroke symptoms, incurred significantly more costs compared to patients who went directly home and reported fewer stroke symptoms. The QoL was stable over time except for stroke patients over 75 years of age, where a significant and clinically meaningful decrease in QoL over time was observed.

In general, studies investigating the economic burden of strokes often do not document non-healthcare costs (e.g., costs related to informal care provision, unpaid help, and productivity losses), despite their significant contribution to the total costs [8,14]. In addition, most burden of disease studies involve top-down methods of costing [15]. Although time-consuming, bottom-up micro-costing is generally seen as the gold-standard method for hospital service costing, leaving the top-down method arguably inferior [35,36]. Based on the findings of a recent systematic review [8], there is a need for long-term (>18 months) post-stroke burden of disease studies. Only three of the included studies [37,38,39] report findings on two-year cost data [8]. The differences in costs of these three studies compared to our study could be explained by methodological discrepancies. For instance, our 24-month estimate of over EUR 47,000 per stroke patient is almost EUR 27,000 (recalculated using purchasing power parity) higher than the Swedish estimate of Ghatnekar et al., 2014 [37]. In the Swedish study, the source for the estimation of resource use (such as rehabilitation, secondary prevention drugs, and productivity losses) was the literature, whereas the Restore4Stroke study used questionnaires. Furthermore, the friction cost method [13] was used to estimate the costs of productivity losses in the Restore4stroke study compared to the human capital approach in the Swedish study [37]. The latter method generates higher costs as it includes every hour not worked due to illness, possibly until the patient’s retirement age. The friction cost method only counts hours lost as time taken until another person assumes the patient’s work [28]. In the two other cost of illness studies reporting two-year data—one performed in Germany [39] (healthcare perspective) and the other in Australia (societal perspective) [38]—the estimated direct medical costs ranged from EUR 13,000 to EUR 17,000 (recalculated using purchasing power parity).However, there are striking differences between the design of each of these studies and the Restore4stroke study. In the German study [39], the cost estimates were based on tariffs, which do not reflect actual costs unlike the real cost prices used in the Restore4Stroke study. The Australian COI study [38] used a top-down approach for costing and claims data for the identification of resource use, which was different to a bottom-up approach for costing and patient-reported resource use in the Restore4Stroke study. These cost differences could be explained by disparities in the study populations and divergences in the provided treatments or clinical practice guidelines for strokes [40,41]. For example, recent changes in stroke care due to the implementation of better acute medical care, such as thrombolysis and thrombectomy, may differ between countries. Finally, cultural differences such as the availability of informal care [42] in a specific country could also have influenced the study findings.

The costs of a stroke have previously been associated with stroke severity in the first-year post-stroke [43,44] and were confirmed in our study. Additionally, we were able to observe this for a long-term follow-up. Furthermore, discharge to rehabilitation clinics and nursing homes was also a significant determinant for higher costs in both short- and long-term follow-ups, indicating the importance of considering this factor for the calculation of total costs in a specific stroke cohort. Our study confirmed the statement of Evers et al. in 2004 [45]: “Stroke puts a substantial burden on the informal caregivers in the first 2 years post-stroke”. In both the first and the second year after stroke, this could be considered high, with an annual average cost of EUR 2000 (150 h). The reason for this is unclear and needs further investigation. It could be related to emotional support or more practical help for stroke patients, including household tasks or transport.

Besides the costs, the QoL was also investigated. For stroke survivors, a QoL utility score of 0.68 to 0.73 was reported in previous research [46,47], which is comparable to the observed two-year utility score of 0.71 in the Restore4Stroke Cohort. However, there is a need to carefully monitor post-stroke patients as they have a lower generic QoL compared to age group norms [48,49,50]. This is especially important for stroke patients in the Restore4Stroke Cohort over 75 years of age; besides a low score in the first 2 years, they also show a clinically relevant decline in the second year post-stroke. A recent systematic review on health state utility values of people with strokes reported the same findings [50]. A stroke aftercare program, which provides emotional support and psycho-education to stroke patients, could focus on this elderly group [48]. 

The strengths of the current study are: (1) the data were obtained through a prospective multi-center observational cohort study entitled ‘’Restore4Stroke’’, of which the design [14,17] was published, and due to strict monitoring of patients missing data were severely limited; (2) the bottom-up costing approach—assisted by trial nurses—is more reliable for resource measurement compared to top-down costing methods [51,52]; (3) the study was performed according to the national guidelines for conducting economic evaluations [19] and guidelines for reporting [20,21]; and (4) the geographical area of the Restore4Stroke Cohort is broad, which could have a positive effect on the generalization of the study findings for the stroke population in the Netherlands. 

This study has the following limitations: (1) the use of retrospective, patient-reported resource use might have led to recall bias [53]; (2) only 18% of the patients suffered from moderate or major stroke symptoms, which limits the validity of our findings for this population; and (3) the generic QoL was measured using EQ-5D-3L, which seems not to be the most optimal measure for stroke patients [29].

Currently, there is a lack of recent data regarding long-term follow-up (>18 months) of BoD studies, using a societal perspective, a prospective design, and a bottom-up costing approach for stroke survivors. Detailed information on the BoD methodology used, the population included, and the treatment given is needed to compare different study findings; it is clear that guidance is highly recommended on how to interpret and report the results of different BoD studies [54].

The data of the BoD studies are also essential for developing reliable model-based economic evaluations, which are needed to estimate the cost-effectiveness of stroke treatments with a lifelong follow-up. These models can be used by policymakers to make better-informed decisions by taking into account both the costs and effects of stroke treatments.

In addition, these studies provide policymakers with better insights into national health reforms in society. The current Dutch policy aims to stimulate the informal care of (stroke) patients, rather than reimbursing paid home care to decrease healthcare spending. This study shows that the hours spent by informal caregivers of stroke patients in the first two years are relatively high. Besides other factors such as the patient’s health-related quality of life and the severity of the stroke, the number of hours informal care received are highly associated with the risk of burnout for caregivers [55].

Finally, more research is needed to investigate the impact of older age (>75 years) on generic QoL ratings in post-stroke patients. For instance, does this group require special monitoring and help to improve their QoL after stroke?

## 5. Conclusions

The average total societal cost for patients two years post-stroke was almost EUR 50,000. The total healthcare cost decreased in the second year compared to the first-year post-stroke due to a decrease in the costs of inpatient care and “rehabilitation treatments” in the second year. However, the healthcare costs in the second year post-stroke could still be considered high. On the other hand, non-healthcare costs have a substantial impact on the total societal costs at 24 months. Home care, informal care, the inability to perform unpaid labor, and productivity losses are all important cost categories to consider for the long-term cost of illness studies. The severity of disease and discharge destination have an impact on total societal costs. The stroke patient scores included in the Restore4Stroke cohort reported a lower QoL than the age group norms but was stable over the first two years post-stroke. Patients over 75 years of age show a clinically meaningful decline in QoL in the first two years post-stroke, which requires further attention.

## Figures and Tables

**Figure 1 ijerph-19-11110-f001:**
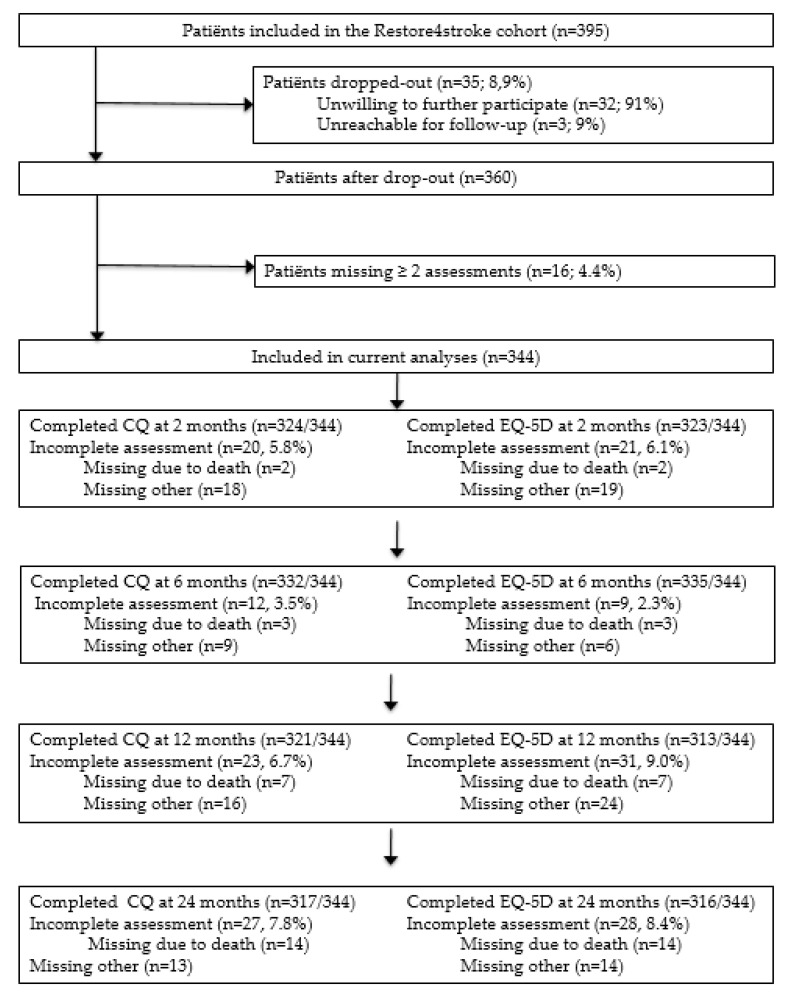
Flowchart for patient inclusion and missing data. Abbreviations: CQ—cost questionnaire; EQ-5D—three-level EuroQoL questionnaire.

**Figure 2 ijerph-19-11110-f002:**
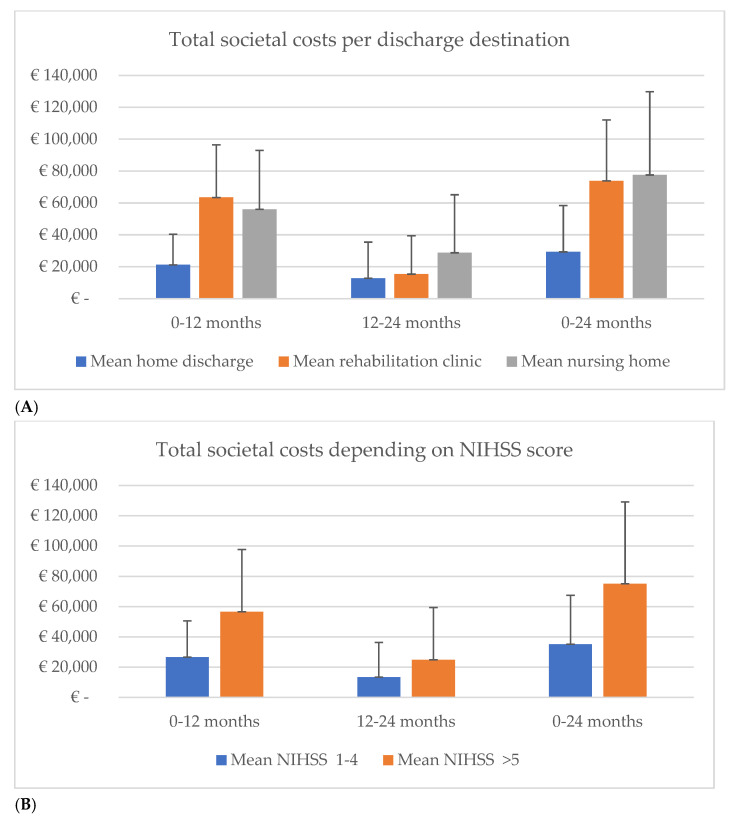
(**A**): subgroup analyses: mean (standard deviation) costs for discharge group, home (*n* = 247, in blue) and rehabilitation clinics (*n* = 48, in red), or Nursing home (*n* = 49, in grey). (**B**): subgroup analyses: mean (standard deviation) costs per NIHSS scores, 1–4 (in blue) and >5 (in red). NIHSS—National Institutes of Health Stroke Scale.

**Table 1 ijerph-19-11110-t001:** Baseline patients’ characteristics (*n* = 344).

	*N*	Mean (SD) or %
Age (in years)	344	66.7 (12.2%)
Gender	344	
Female	121	35.2%
Male	223	64.8%
Marital status	344	
Living together	240	69.8%
No relationship	104	30.2%
Education	341	
Low	250	73.3%
High	91	26.7%
Stroke type	343	
Ischaemic stroke	319	92.7%
Hemorrhagic stroke/Infarction stroke	24	7.0%
Severity of stroke (NIHSS)	344	
No stroke symptoms (NIHSS 0)	84	24.4%
Minor stroke symptoms (NIHSS 1–4)	197	57.3%
Moderate stroke symptoms (NIHSS 5–12)	58	16.8%
Moderate to severe stroke symptoms (NIHSS ≥ 13)	5	1.5%
Residence after discharge	344	
Home	247	71.7%
Rehabilitation center	48	14.0%
Geriatric rehabilitation	49	14.2%

SD—standard deviation; NIHSS—National Institutes of Health Stroke Scale.

**Table 2 ijerph-19-11110-t002:** Total resource use and costs (bootstrapped) in the first two years post-stroke (*n* = 344).

		Users		Resource Use Per Patient		Costs Per Patient		
	Unit	*N*	%	Mean	SD	Mean	SD	Median
**Healthcare costs**								
General practitioner	Contact	332	96.5	13.7	16.79	€481	€34	€478
Specialist	Contact	335	97.4	11.2	9.45	€1335	€65	€1333
Allied health professionals	Contact	269	78.2	32.2	46.33	€1105	€93	€1104
Mental healthcare professionals	Contact	125	36.3	1.9	5.93	€197	€35	€195
Rehabilitation treatments	Day	261	75.9	29.9	41.31	€8664	€673	€8645
Hospital	Night	307	89.2	10.2	11.39	€6864	€448	€6851
Rehabilitation clinic	Night	130	37.8	13.0	29.66	€6301	€822	€6279
Nursing home	Night	94	27.3	5.9	36.40	€1038	€351	€1012
Psychiatric clinic	Night	61	17.7	0.4	3.67	€133	€69	€127
Medication	Various	-	-	-	-	€1119	€53	€1118
**Total healthcare costs**						€27,159	€1611	€27,174
**Non-healthcare costs**								
Paid home care	Hours	153	44.5	185.7	569.81	€3979	€695	€3925
Informal care	Hours	238	69.2	291.5	598.66	€4320	€512	€4298
Inability to perform unpaid labor	Day	216	62.8	69.0	130.96	€8284	€895	€8241
Productivity Losses *	Day	41	11.9	22.0	76.01	€3810	€406	€3791
**Total non-healthcare costs**						€20,330	€1603	€20,302
**Total societal costs**						€47,502	€2628	€47,384

SD—standard deviation, * estimated by the friction cost method.

**Table 3 ijerph-19-11110-t003:** Resource use and costs (bootstrapped) for the first year and the second year post-stroke (*n* = 344).

Unit		0–12 Months Post-Stroke					12–24 Months Post-Stroke					Difference
Per Patient		Resource Use		Cost			Resource Use		Cost			
		Mean	SD	Mean	SD	Median	Mean	SD	Mean	SD	Median	Mean (95% CI) *
Healthcare Costs												
GP	Contact	6.9	5.00	€237	€9.68	€236	6.9	15.60	€247	€32	€243	€10 (−€44, €87)
Specialist	Contact	8.1	6.73	€946	€44.67	€947	3.2	5.11	€391	€37	€389	−€556 (−€667, −€437)
Allied HP	Contact	21.2	28.03	€718	€55.13	€717	11.0	27.38	€386	€56	€385	−€332 (−€487, −€181)
Mental HP	Contact	1.3	4.12	€136	€23.92	€135	0.6	3.15	€61	€19	€59	−€75 (−€138, −€15)
Rehabilitation treatments	Day	22.9	29.86	€6582	€480.22	€6595	7.0	23.49	€2064	€387	€2050	−€4518 (−€5,768, −€3226)
Hospital	Night	8.4	8.68	€5601	€336.73	€5587	1.8	6.74	€1241	€272	€1233	−€4360 (−€5233, −€3494)
Rehabilitation clinic	Night	10.9	25.68	€5205	€713.72	€5164	2.1	15.31	€1089	€455	€1033	−€4116 (−€5761, −€2478)
Nursing home	Night	4.0	18.91	€711	€195.33	€692	1.9	20.16	€346	€208	€334	−€365 (−€879, €210)
Psychiatric clinic	Night	0.1	0.70	€28	€12.41	€26	0.3	3.54	€102	€65	€98	€74 (−€14, €236)
Medication	Various	-	-	€531	€23.77	€531	-	-	€536	€38	€533	€5 (−€80, €96)
**Total healthcare costs**				€20,709	€1282	€20,639	-	-	€6431	€846	€6406	−€14,277 (−€17,319 −€11,236)
Paid home care	Hours	97.4	349.47	€2056	€430	€2023	88.3	385.73	€1878	€459	€1836	−€178 (−€1431, €1042)
Informal care	Hours	159.7	359.73	€2323	€290	€2304	131.8	423.28	€1996	€378	€1970	−€326 (−€1228, €660)
Inability to perform unpaid labor	Day	27.4	50.98	€3186	€347	€3180	41.7	106.17	€5068	€744	€5081	€1882 (€327, €3478)
Productivity Losses *	Day	22.0	76.01	€3819	€389	€3807	-	-	0	0	0	−€3819 (−€4558, −€3065)
**Total non-healthcare costs**				€11,336	€817	€11,324			€8910	€1078	€8874	−€2426 (−€5079, €246)
**Total societal costs**				€32,085	€1743	€31,984			€15,383	€1526	€15,372	−€16,703 (−€21,243, −€12,039)

Abbreviations: GP—general practitioner, HP—healthcare professionals, SD—standard deviation, NA—Not applicable, 95%CI—95% confidence interval. * estimated by the friction cost method. All significant differences are highlighted in red.

**Table 4 ijerph-19-11110-t004:** EQ-5D-3L dimensions and Dutch utilities at 12 months and 24 months post-stroke (*n* = 344).

	*N*	12 Months Post-Stroke	24 Months Post-Stroke	
Dimensions EuroQoL		mean (SD)	mean (SD)	Mean difference * (95%CI)
Mobility	344	1.54 (0.59)	1.60 (0.58)	0.06 (−0.03, 0.15)
Self-care	344	1.23 (0.52)	1.30 (0.59)	0.07 (−0.02, 0.16)
Usual activities	344	1.67 (0.70)	1.63 (0.69)	−0.04 (−0.15, 0.07)
Pain/discomfort	344	1.56 (0.59)	1.64 (0.64)	0.08 (−0.02, 0.17)
Anxiety/depression	344	1.38 (0.57)	1.45 (0.62)	0.06 (−0.03, 0.16)
Average utility score	344	0.7400 (0.2662)	0.7094 (0.3153)	−0.03 (−0.08, 0.01)
Utility score EuroQoL: Age				
<65	146	0.7611 (0.2546)	0.7688 (0.2667)	0.01 (−0.05, 0.07)
65–75	99	0.7023 (0.3060)	0.7277 (0.2894)	0.03 (−0.06, 0.10)
>75	99	0.7467 (0.2378)	0.6036 (0.37714)	−0.14 (−0.23, −0.05)
Utility score EuroQoL: Gender				
Male	223	0.7497 (0.2619)	0.7290 (0.3182)	−0.02 (−0.07, 0.03)
Female	121	0.7222 (0.2744)	0.6734 (0.3080)	−0.05 (−0.12,0.02)
Utility score EuroQoL: Severity of stroke				
No stroke symptoms (NIHSS 0)	84	0.7305 (0.2782)	0.7076 (0.3414)	0.02 (−0.07, 0.11)
Minor stroke symptoms (NIHSS 1–4)	197	0.7352 (0.2519)	0.7280 (0.2892)	−0.01 (−0.06, 0.04)
Moderate stroke symptoms (NIHSS 5–12)	58	0.7713 (0.2918)	0.6501 (0.3550)	−0.12 (−0.23, 0.01)
Moderate to severe stroke symptoms (NIHSS ≥ 13)	5	0.7302 (0.3668)	0.6973 (0.37659)	−0.03 (−0.42, 0.38)

* Bootstrapped mean difference, 95%CI: 95% confidence interval, NIHSS—National Institutes of Health Stroke Scale. All significant differences are highlighted in red.

**Table 5 ijerph-19-11110-t005:** Sensitivity and Subgroup analyses.

Sensitivity Analyses Costs		Base Case *		Sensitivity Analyses			
	Costs	Mean Total Costs (€)	SD	Mean Total Costs (€)	SD	Mean Difference	95% CI
Method extrapolation costs	(18–24M)*2)/ (6–12M) + (18–24M)	€42,378	€2225	€46,282	€2597	€3904	(€−2751, €11,146)
Perspective	(Societal/Healthcare)	€42,506	€2389	€27,125	€1620	€−15,382	(€−21,244, €−9785)
**Sensitivity analyses Quality of life**		**Utility**	**SD**	**Utility**	**SD**	**Mean difference**	**95% CI**
Average utility 12M	Dutch/UK	0.7110	0.0185	0.6604	0.0215	−0.0506	(−0.1104, 0.0081)
Average utility 24M	Dutch/UK	0.7413	0.0156	0.6769	0.0192	−0.0644	(−0.1122, −0.0167)
**Subgroup analysis of quality of life**		**12 months post-stroke**			**24 months post-stroke**		
Characteristics	Group	Mean difference (utility)	95% CI		Mean difference (utility)	95% CI	
Gender	Male/Female	−0.0277	(−0.0883, 0.0301)		−0.0559	(−0.1262, 0.0111)	
Age	65+/65−	−0.0361	(−0.0932, 0.0193)		0.1030	(0.0395, 0.1664)	
Stroke type	Infarction/Haemorrhage	−0.1245	(−0.2638, 0.0058)		0.0557	(−0.0459, 0.1388)	
Recurrent stroke	Yes/No	−0.0048	(−0.0941, 0.0913)		0.0502	(−0.0610, 0.1699)	
Education	High/Low	0.0198	(−0.0473, 0.0910)		0.0118	(−0.0722, 0.0901)	
Home Discharge	Yes/No	−0.0292	(−0.0960, 0.0351)		−0.1030	(−0.1796, 0.0298)	
Stroke severity	0–4/>5	0.0343	(−0.0426, 0.1100)		−0.0695	(−0.1666, 0.0147)	
**Subgroup analysis costs**		**0-12 months post-stroke**		**12-24 months post-stroke**		**0-24 months post-stroke**	
Characteristic	Group	Mean difference (€)	95% CI	Mean difference (€)	95% CI	Mean difference (€)	95% CI
Gender	Male/Female	−€1280	(−€7583, €5729)	€3897	(−€1620, €9587)	€2352	(−€5555, €10,872)
Age	65+/65−	−€3148	(−€9339, €2894)	€4435	(−€945, €9583)	€2224	(−€6147, €10,050)
Stroke type	Infarction/Haemorrhage	€3323	(−€10,441, €18,298)	€2806	(−€9427, €18,949)	€7114	(−€13,288, €31,104)
Recurrent stroke	Yes/No	€4631	(−€3222, €11,567)	€5596	(−€2933, €14,694)	−€814	(−€11,490, €11,589)
Education	High/Low	€2870	(−€4068, €9835)	€3276	(−€2029, €8888)	€5311	(−€4122, €14,389)
Home Discharge	Yes/No	€38,535	(€31,117, €45,654)	€9347	(€2602, €16,283)	€46,503	(€36,877, €56,409)
Stroke severity	0–4/>5	€29,971	(€19,768, €40,477)	€11,545	(€2993, €20,705)	€40,144	(€26,561, €54,317)

***** Base case analysis uses the original cost exploration method, it uses the societal perspective to calculate total costs and calculates utilities with a Dutch tariff. Sensitivity analyses use another extrapolation method to calculate the costs for the second year post-stroke, estimates of total healthcare costs and utilities are calculated with a UK tariff. All significant differences are highlighted in red. Abbreviations: M—months, SD—standard deviation, UK—United Kingdom, and 95%CI—confidence interval, Stroke severity—National Institutes of Health Stroke Scale.

## Data Availability

The datasets used and/or analyzed during the current study are available from the corresponding author upon reasonable request.

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
