# Peer review of "Estimating the Burden of Stroke: Two-Year Societal Costs and Generic Health-Related Quality of Life of the Restore4Stroke Cohort"

_ijerph, 2022, doi:10.3390/ijerph191711110_

Round 1

Reviewer 1 Report

       This study investigated two-year societal costs and generic health-20 related quality of life (QoL) using a bottom-up approach for the Restore4stroke cohort, this topic is of great interest. Overall, the methodology is clearly presented, below are some granular comments:

       Why were some results heighted in red? Any special meaning?        In terms of handling missing data, multiple imputations were applied to replace all missing data using the following 172 predictors: gender, marital status, age, treatment location and severity of stroke, when using multiple imputation, were missing values replaced by the mean for those variables? However, these assumptions are often unrealistic, will this approach potentially cause any bias?         Why did the Bootstrapping of cost data use 1000 replications? Was this sufficient?        Costs were mainly compared and interpreted based on the mean values, how about the median values? Did the median values also display the same message?

Author Response

Dear reviewer,

thank you very much for reviewing our paper.

In the attachted document you can find the answers to your questions.

Kind regards,

Ghislaine van Mastrigt

ps we also included the answers for other reviewer in this document

Reviewer 2 Report

This study aimed to investigate two-year societal costs and generic health-related quality of life (QoL) in patient with stroke using a bottom-up approach for the Restore4stroke cohort. This study found that the non-healthcare costs have a substantial impact on the first and second year total societal costs post-stroke. The current study suggests that a societal perspective with at least a follow-up of two years is highly recommended to get a complete picture of all relevant costs related to stroke. Additionally, more research is needed to investigate the decline in QoL found in stroke patients above the age of 75 years. The authors wrote the study results and conclusions well using appropriate statistical methods. Here are my suggestions:

1.       There are some typos.

2.       In Figure 1, the text is small and underlined, making it difficult to read. Also, the description of the abbreviation is not described in the legend.

3.       In Table 1, among stroke types, ischemic stroke and infarct stroke have the same meaning. Could a hemorrhagic stroke be incorrectly described as an infarct stroke?

4.       What does “GP” mean in the text?

5.       In Methods, it seems necessary to describe what high or low EQ-5D-3L scores mean on quality of life. For example, the lower the score, the worse the quality of life.

Author Response

(The authors gave the same response as above.)
